# Kernel graph filtering—A new method for dynamic sinogram denoising

**Shiyao Guo**[1], **Yuxia Sheng**[1]*, **Li Chai**[1], **Jingxin Zhang**[2]

**1** Engineering Research Center of Metallurgical Automation and Measurement Technology, Wuhan University of Science and Technology, Wuhan, China, **2** School of Science, Computing and Engineering Technology, Swinburne University of Technology Melbourne, VIC, Australia

\* shengyuxia@wust.edu.cn

## Abstract

Low count PET (positron emission tomography) imaging is often desirable in clinical diagnosis and biomedical research, but its images are generally very noisy, due to the very weak signals in the sinograms used in image reconstruction. To address this issue, this paper presents a novel kernel graph filtering method for dynamic PET sinogram denoising. This method is derived from treating the dynamic sinograms as the signals on a graph, and learning the graph adaptively from the kernel principal components of the sinograms to construct a lowpass kernel graph spectrum filter. The kernel graph filter thus obtained is then used to filter the original sinogram time frames to obtain the denoised sinograms for PET image reconstruction. Extensive tests and comparisons on the simulated and real life in-vivo dynamic PET datasets show that the proposed method outperforms the existing methods in sinogram denoising and image enhancement of dynamic PET at all count levels, especially at low count, with a great potential in real life applications of dynamic PET imaging.

## Introduction

Positron emission tomography (PET) is a functional imaging modality. By monitoring the distribution of radioactive tracers, it can detect the early onset of diseases [1], such as Alzheimer's disease. To show the rapid change of in-vivo radioactive tracer accurately, time frame with short scan durations should be applied, especially in early frame. This however leads to lower counts in each frame, which results in lower signal-to-noise ratio of reconstructed PET image. To improve the quality of PET image with low counts, three types of methods have been proposed in the literature.

The first type attempts to improve image reconstruction. Its representative is the reconstruction algorithms incorporating prior information [2]. The prior information used includes anatomical information [3, 4]. Compared to the reconstruction methods without using priors, these methods can significantly reduce the noise level of PET image. However, these methods generally increase computational cost and may introduce artifacts. For example, when magnetic resonance (MR) images are used as prior information in PET image reconstruction, some MR-only information, such as the bone or lesion existing only in MR image [4], may be introduced in the reconstructed PET image.

**Data Availability Statement:** The simulation data are not third party data and there are no restrictions. We use brain model provided in Simulated Brain Database (https://brainweb.bic.mni.mcgill.ca/) to generate simulation data. Details on how to generate the data are given in our paper.

We also provide the data as Supporting information files (PET_data.xlsx).

**Funding:** This work was supported by National Natural Science Foundation of China under Grant Nos.61625305 and 61501337.

**Competing interests:** The authors have declared that no competing interests exist.

The second type of methods attempts to post-denoise the reconstructed PET image. Among these methods, Gaussian filtering is a simple and traditional method for reducing the noise in PET image, but it may lead to blurring of image edges. In order to produce PET image with better details and lower noise, various denoising filters, such as bilateral filter [5], non-local means (NLM) filter [6, 7] and graph filter [8] have been proposed. Although these methods have good denoising performance for PET image with high counts, for image with lower counts, noise in image domain is highly correlated and is very difficult to reduce.

PET image is reconstructed from the sinogram data acquired from PET scanning process. The third type of methods therefore attempts to pre-denoise sinogram data before image reconstruction [9]. Since sinograms can be well modeled using Poisson random variables [10–12], noise in projection domain is relatively easier than in image domain. The noise in sinograms can affect all pixels in reconstructed image during reconstruction process. These make the noise in PET image more complex and difficult to suppress, and stimulate the development of sinogram denoising. It has been shown that pre-denoising in sinograms is more efficient than post-denoising in PET images [13], since pre-denoising slightly aligns neighbouring projections in the sinogram. A number of sinogram pre-denoising methods have been proposed in the literature, including the popular methods of block matching 3-D (BM3D) [14] and sinogram-based dynamic image guided filtering (SDIGF) [15]. These methods perform relatively well, but their effectiveness in very low count sinograms is limited. This is because the noise corruption in low count sinograms is highly complex, and the denoising methods based on some conventional assumptions may not effectively reduce the noise.

Graph signal processing (GSP) is a new paradigm recently emerged in the signal processing field for data processing. The core of GSP is to use graph to describe the relations between data points and treat the data as the signals on the vertices (nodes) of the graph. Based on this concept, a range of GSP techniques, such as graph Fourier transform, graph spectrum analysis and filtering, GSP based graph learning and deep learning on graph, have been developed in recent years. These techniques have been used in various signal and image processing [16], segmentation [17] and classification [18] problems, resulting in many exciting new results [19, 20].

Inspired by the success of GSP, we present in this paper a novel graph filtering method for dynamic sinogram denoising to significantly improve PET image quality. We treat the dynamic sinograms as the signals on a graph, and learn the graph adaptively from the kernel principal components of the sinograms to construct a lowpass kernel graph spectrum filter. The kernel graph filter thus obtained is then used to filter the original sinograms to obtain the denoised sinograms for PET image reconstruction. We further use the simulated and real life in-vivo datasets to validate and evaluate the performance of the proposed new method, and demonstrate its excellent performance and advantages over the existing methods by comprehensive comparisons.

The paper is organized as follows. Preliminaries section briefly introduces the dynamic PET sinograms and GSP. Methods section describes the proposed method and its derivation in detail. Result section presents the simulation and real life in-vivo studies to compare the proposed method with other methods. Finally, the Discussion and Conclusion sections are presented in the end of paper.

## Preliminaries

Throughout the paper, boldface letters represent matrices and vectors, eg a matrix $\boldsymbol{A}$ and a vector $\boldsymbol{x}$, lower case letters represent scalars (including scalar elements of matrices, vectors and sets), eg an eigenvalue $\lambda$ and the $ij$th element $a_{ij}$ of the matrix $\boldsymbol{A}$, and boldface calligraphic letters represent sets, eg the set of $M \times M$-dimensional real valued matrices $\mathcal{R}^{M \times M}$.

## Noisy dynamic sinograms

Let $\boldsymbol{p}_1, \cdots, \boldsymbol{p}_i, \cdots, \boldsymbol{p}_N \in \mathcal{R}^M$ be the noisy sinograms from dynamic PET imaging, where $\boldsymbol{p}_i = [p_{i1}, p_{i2}, \cdots, p_{iM}]^T$ are the noisy sinograms for the $i$th frame of PET activity image and $M$ is the total number of lines of response for all angles. Let $\boldsymbol{P} = [\boldsymbol{p}_1, \cdots, \boldsymbol{p}_i, \cdots, \boldsymbol{p}_N] \in \mathcal{R}^{M \times N}$ be the data matrix consisting of $N$ frames of noisy sinograms. Here, the sinogram frames $\boldsymbol{p}_i$'s are ordered and indexed by the occurrence of their corresponding PET activity time intervals. The activity time interval of $\boldsymbol{p}_i$ precedes that of $\boldsymbol{p}_{i+1}$, and the sinogram data $\boldsymbol{p}_i$ are acquired before the sinogram data $\boldsymbol{p}_{i+1}$ during PET scan process. Hence, a smaller $i$ represents an earlier time interval and a larger $i$ represents a later time interval in the scan.

According to dynamic PET physics, at the beginning of PET scan, namely, at the early frames with small $i$, the tracer concentration of blood is high, and the initial tracer concentration in interstitial and intracellular space of tissue is very low and increases rapidly. As a result, the signal energy of $\boldsymbol{p}_i$ is low and grows rapidly when $i$ is small. To capture the rapid changes of energy in early frames, shorter time intervals are usually used to collect the sinograms $\boldsymbol{p}_i$ when $i$ is small. In late frames, most of the tracer resides in intracellular space, and the tissue energy becomes high and saturated. To capture the slower changes at higher energy level in later frames, the time intervals are gradually increased, resulting in higher signal energy of $\boldsymbol{p}_i$ when $i$ is large. Hence, when measured by the 1-norm, the energy of $\boldsymbol{p}_i$, $\|\boldsymbol{p}_i\|_1$, satisfies $0 < \|\boldsymbol{p}_i\|_1 \leq \|\boldsymbol{p}_{i+1}\|_1 \leq \|\boldsymbol{p}_N\|_1$ for $1 \leq i \leq N-1$. See Fig 1(b) in Result Section for illustration. Physically, these mean that the sinograms of early frames, ie $\boldsymbol{p}_i$'s with small $i$, have lower counts and substantially higher noise [6] as compared to the sinograms of late frames, ie $\boldsymbol{p}_i$'s with large $i$, which have higher counts and significantly lower noise.

Dynamic PET sinograms are acquired from the same anatomy using the same scanner. Although they represent PET activities of the same region at different time intervals, they all show similar structures due to the same underlying anatomy. Therefore, the clearer structural information from later sinogram frames with lower noise can be used to reduce the noise of earlier sinogram frames by filtering along the temporal direction across all sinogram frames. This idea is exploited in this paper to develop a new sinogram denoising method.

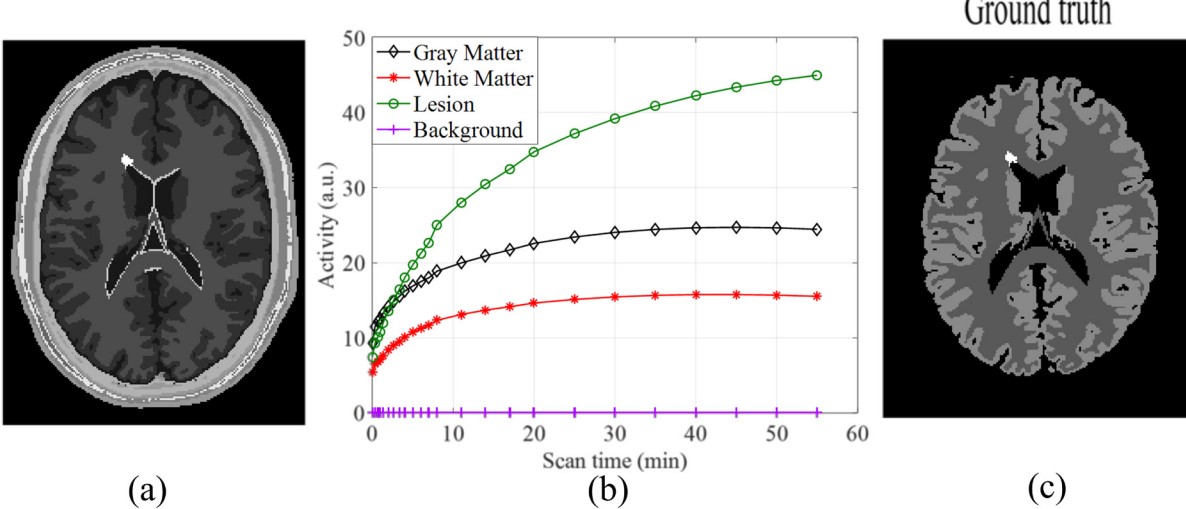

(a)  (b)  (c)

**Fig 1.** (a) Anatomical model, (b) time activity curves, (c) simulated PET activity image at the end of scan. The bright spot in (a) and (c) is a lesion simulated and inserted by BrainWeb.

The sinogram denoising method to be proposed is based on data adaptive graph filtering and its derivation does not rely on any specific assumption about the noise in the noisy sinograms $\boldsymbol{p}_i$'s.

**Graph signal processing.** Let $G(\mathcal{V}, \mathcal{E}, \boldsymbol{W})$ be a weighted and undirected graph with a set of vertices $\mathcal{V} = \{v_1, v_2, .., v_N\}$, a set of edges $\mathcal{E} \subseteq \mathcal{V} \times \mathcal{V}$ and a weighted adjacency matrix $\boldsymbol{W} \in \mathcal{R}^{N \times N}$. When the vertices $v_i$ and $v_j$ are connected by an edge $e_{ij} \in \mathcal{E}$, the $ij$th element of $\boldsymbol{W}$, denoted as $w_{ij}$, is nonzero and equals the weight of the edge, otherwise $w_{ij}$ is zero. Define $d_i := \sum_{j=1}^{N} w_{ij}$ as the degree of vertex $v_i$, and $\boldsymbol{D} := diag\{d_1, d_2, \cdots, d_N\}$ as the graph degree matrix. The graph Laplacian matrix is given as $\boldsymbol{L} = \boldsymbol{D} - \boldsymbol{W}$, which is symmetric and positive semi-definite. The eigenvalues of $\boldsymbol{L}$ are real valued in an ascending order $0 \leq \lambda_1 \leq \lambda_2 \leq \cdots \leq \lambda_n$ and are defined as the spectra of the graph $G$ [16].

A signal defined on a graph is called a graph signal. On a graph $G$ with $N$ vertices, a graph signal can be represented by a vector $\boldsymbol{x} \in \mathcal{R}^{1 \times N}$ consisting of the data from the $N$ vertices of $G$ [16]. A simple example is the vector consisting of the intensity values of $N$ pixels in an image modeled by a graph $G$, with its $N$ vertices representing the $N$ pixels. For a graph signal $\boldsymbol{x} \in \mathcal{R}^{1 \times N}$, an operator $\boldsymbol{F} \in \mathcal{R}^{N \times N}$ that yields $\boldsymbol{y} = \mathbf{xF}$ represents a graph filter with the filter output $\boldsymbol{y} \in \mathcal{R}^{1 \times N}$. The adjacency matrix $\boldsymbol{W}$ and the Laplacian matrix $\boldsymbol{L}$ of a graph $G$ are two such operators with the lowpass and highpass natures, respectively, in the graph spectrum domain [21].

For a graph $G$, its adjacency matrix $\boldsymbol{W}$ can be found by using a kernel function and the samples of the graph signal $\boldsymbol{x}$ defined on $G$ [21]. See Method section for details.

## Methods

### The proposed method–kernel graph filtering

The sinograms from dynamic PET scan process are always noisy, especially in the low count frames. This seriously affects the performance of all PET image reconstruction methods. Noise corruption in the low count sinograms is highly complex, classical methods based on some conventional assumptions may not effectively reduce the noise. To overcome this difficulty, we propose a novel kernel graph filtering method to denoise the dynamic sinograms for dynamic PET image reconstruction.

We first use kernel principal component analysis (PCA) [22] to extract the kernel principal components of sinograms and hence reduce their dimension. Then, we use the low dimensional kernel principal components of sinograms to construct the graph filter and use the filter to denoise sinograms. Finally, we use the denoised sinograms in two popular PET image reconstruction algorithms to reconstruct PET images.

Presented below are these three steps, followed by four remarks on their justification and relation to previous work.

Step 1: *Kernel principal component extraction of sinograms*

Given the noisy dynamic sinograms $\boldsymbol{p}_1, \cdots, \boldsymbol{p}_i, \cdots, \boldsymbol{p}_N \in \mathcal{R}^M$, we construct the kernel matrix $\boldsymbol{C} = [c_{ij}]$ using the radial Gaussian kernel function [23, 24]

$$c_{ij} = exp(- \parallel \boldsymbol{p}_i - \boldsymbol{p}_j \parallel^2 / 2\sigma_1^2) = \langle \phi(\boldsymbol{p}_i), \phi(\boldsymbol{p}_j) \rangle, \ \sigma_1 > 0, \ i, j = 1, 2, \cdots, N. \tag{1}$$

where $\phi(\cdot)$ is an underlying nonlinear function that implicitly maps the noisy sinograms to the infinite dimensional feature vectors $\phi(\boldsymbol{p}_1), \cdots, \phi(\boldsymbol{p}_i), \cdots, \phi(\boldsymbol{p}_N)$, and manifests the underlying features of sinograms in the feature space. The $\sigma_1$ in (1) is a free parameter that can be tuned if needed to adjust the width of the kernel.

Because the feature vectors $\phi(\boldsymbol{p}_i)$ are generally not zero mean, we convert the kernel matrix $\boldsymbol{C}$ constructed above to the centered kernel matrix $\hat{\boldsymbol{C}}$

$$\hat{\boldsymbol{C}} = \boldsymbol{C} - \boldsymbol{1}_N\boldsymbol{C} - \boldsymbol{C}\boldsymbol{1}_N + \boldsymbol{1}_N\boldsymbol{C}\boldsymbol{1}_N \tag{2}$$

where $\boldsymbol{1}_N$ is an $N \times N$ matrix with all elements being $1/N$. We then solve $\hat{\boldsymbol{C}}\boldsymbol{\alpha}_l = \lambda_l N\boldsymbol{\alpha}_l$, $l = 1$, $2, \cdots, N$, to find the $N$ eigenvectors of $\hat{\boldsymbol{C}}$

$$\boldsymbol{\alpha}_l = [\alpha_{l1}, \cdots, \alpha_{lj}, \cdots, \alpha_{lN}]^T, \quad l = 1, 2, \cdots, N. \tag{3}$$

Using the centered kernel matrix $\hat{\boldsymbol{C}}$ and its first $d$ eigenvectors $\boldsymbol{\alpha}_l$, $l = 1, 2, \cdots, d$, $d \leq N$, we perform kernel PCA [22] to extract the $d$ principal components of the feature vectors $\phi(\boldsymbol{p}_1), \cdots, \phi(\boldsymbol{p}_i), \cdots, \phi(\boldsymbol{p}_N)$ by the following algorithm.

For each feature vector $\phi(\boldsymbol{p}_i)$, $i = 1, 2, \cdots, N$, compute its $d$ principal components, $y_{li}$, $l = 1, 2, \cdots, d$, and $d \leq N$,

$$y_{li} = \sum_{j=1}^{N}\alpha_{lj}\hat{c}_{ij}, \quad l = 1, 2, \cdots, d, \quad i = 1, 2, \cdots, N, \tag{4}$$

where $\alpha_{lj}$ is the $j$th element of the eigenvector $\boldsymbol{\alpha}_l$ of $\hat{C}$ given in (3), and $\hat{c}_{ij}$ is the $ij$th element of $\hat{C}$. The computation of $y_{li}$ over $l = 1, 2, \cdots, d$ gives the principal component vector $\boldsymbol{y}_i = [y_{1i}, y_{2i}, \cdots, y_{di}]^T \in \mathcal{R}^d$ for $\phi(\boldsymbol{p}_i)$, and the further computation of $y_{li}$ over $i = 1, 2, \cdots, N$ gives the principal component vectors $\boldsymbol{y}_1, \boldsymbol{y}_2, \cdots, \boldsymbol{y}_N$ for $\phi(\boldsymbol{p}_1), \phi(\boldsymbol{p}_2), \cdots, \phi(\boldsymbol{p}_N)$, respectively. The number of principal components $d \leq N$ can be tuned to the given sinogram data.

The above algorithm reduces the infinite dimensional feature vectors $\phi(\boldsymbol{p}_1), \phi(\boldsymbol{p}_2), \cdots, \phi(\boldsymbol{p}_N)$ to the $d$-dimensional kernel principal component vectors $\boldsymbol{y}_1, \boldsymbol{y}_2, \cdots, \boldsymbol{y}_N$, which will be used in Step 2 to construct kernel graph filter.

Step 2: *Kernel graph filter construction and kernel graph filtering of sinograms*

We view the kernel principal component vectors $\boldsymbol{y}_i$, $i = 1, 2, \cdots, N$, obtained in Step 1 as the samples of the graph signal defined on the $N$ vertices of an underlying graph, and use them to obtain the adjacency matrix of the graph by Gaussian kernel function, $\boldsymbol{A} = [a_{ij}]$.

$$a_{ij} = \begin{cases} exp(- \parallel \boldsymbol{y}_i - \boldsymbol{y}_j \parallel^2 /2\sigma_2^2), & \boldsymbol{y}_j \in \mathrm{knn}_i \\ 0, & \mathrm{otherwise} \end{cases}, \quad \sigma_2 > 0, \quad i = 1, 2, \cdots, N, \tag{5}$$

where $\mathrm{knn}_i$ is the $k$-nearest neighbors of $\boldsymbol{y}_i$ [25] and $\sigma_2$ is a free parameter that is tuned in practice to adjust the width of the kernel. The number of the nearest neighbors in $\mathrm{knn}_i$ is determined by

$$k_i = r(N \parallel \boldsymbol{p}_i \parallel_1 / \parallel \boldsymbol{p}_N \parallel_1), \quad i = 1, 2, \cdots, N, \tag{6}$$

where $\|\boldsymbol{p}_i\|_1$ is the 1-norm of the sinogram vector $\boldsymbol{p}_i$, and $r(\cdot)$ is the operator of rounding to the nearest integer.

The adjacency matrix $\boldsymbol{A} = [a_{ij}]$ obtained from (5) defines a graph of $\boldsymbol{y}_i$'s. It is also a kernel graph for the original sinograms $\boldsymbol{p}_i$'s, since $\boldsymbol{y}_i$'s are the kernel principal components of $\boldsymbol{p}_i$'s. A lowpass kernel graph filter can be derived from the adjacency matrix $\boldsymbol{A}$ as described below.

Column normalizing $\boldsymbol{A} = [a_{ij}]$ gives a Markov transition matrix $\boldsymbol{F} = [F_{ij}]$, $i, j = 1, 2, \cdots, N$, with

$$F_{ij} = a_{ij}/\sum_{j=1}^{N}a_{ij}, \quad \sum_{i=1}^{N}F_{ij} = 1. \tag{7}$$

The columns of this Markov transition matrix $\boldsymbol{F} \in \mathcal{R}^{N \times N}$ represent the probability distributions of transition from a particular frame of sinograms to every other frames of sinograms in one step along the column directions of the kernel graph. The $m$th power of $\boldsymbol{F}$, $\boldsymbol{F}^m$, can be used as a low pass graph spectrum filter defined on the kernel graph, with the order $m \geq 1$. The operation

$$\boldsymbol{PF}^m = [\boldsymbol{p}_1, \cdots, \boldsymbol{p}_i, \cdots, \boldsymbol{p}_N]\boldsymbol{F}^m =: [\hat{\boldsymbol{p}}_1, \cdots, \hat{\boldsymbol{p}}_i, \cdots, \hat{\boldsymbol{p}}_N] = \hat{\boldsymbol{P}} \tag{8}$$

yields $\hat{\boldsymbol{p}}_i$'s which are the weighted averages of the original sinograms $\boldsymbol{p}_i$'s along the column directions of the kernel graph. In practice, to find suitable order, the optimal order $m^*$ can be determined by

$$\frac{\parallel \boldsymbol{PF}^{m^*} - \boldsymbol{PF}^{(m^*-1)} \parallel_F^2}{\parallel \boldsymbol{PF}^{(m^*-1)} \parallel_F^2} \leq \epsilon \tag{9}$$

to obtain the optimal kernel graph filter $\boldsymbol{F}^{m^*}$, where $\epsilon > 0$ is a small constant and $\parallel \cdot \parallel_F$ is the Frobenius norm of a matrix.

Filtering the sinogram data $\boldsymbol{P}$ with the optimal kernel graph filter $\boldsymbol{F}^{m^*}$ obtained above gives

$$\hat{\boldsymbol{P}}^* = \boldsymbol{PF}^{m^*}. \tag{10}$$

This is the denoised sinograms of the proposed kernel graph filtering (KGF) method for dynamic PET sinograms.

Step 3: *PET image reconstruction with kernel graph filtered sinograms*

The kernel graph filtered sinograms $\hat{\boldsymbol{P}}^*$ can be used as input data in any existing dynamic PET image reconstruction methods to improve their image quality. In this step, we only use $\hat{\boldsymbol{P}}^*$ in two popular iterative reconstruction methods: the traditional method of Maximum-Likelihood Expectation Maximization (MLEM) [26, 27] and the recent method of Kernalized Expectation Maximization (KEM) [10]. The intention is to show the image quality improvement from using $\hat{\boldsymbol{P}}^*$, instead of proposing a new method for dynamic PET image reconstruction.

Both MLEM and KEM methods treat the sinograms as the output of PET forward projection. Following the same line, we treat the kernel graph filtered sinogram $\hat{\boldsymbol{P}}^*$ as the output of PET forward projection

$$\hat{\boldsymbol{P}}^* = \boldsymbol{HX} + \boldsymbol{S}, \tag{11}$$

where $\boldsymbol{H} \in R^{M \times M_1}$ is the systems matrix, $\boldsymbol{X} = [\boldsymbol{x}_1, \cdots \boldsymbol{x}_i, \cdots, \boldsymbol{x}_N] \in R^{M_1 \times N}$ is the matrix of $N$ frames of unknown dynamic PET images to be reconstructed, with $\boldsymbol{x}_i = [x_{i1}, x_{i2}, \cdots, x_{iM_1}]^T$ the vector consisting of the column stacked image pixels of the $i$th frame, and $\boldsymbol{S} \in R^{M \times N}$ is the expectation of random and scattered events [10, 26]. Hence, we can use $\hat{\boldsymbol{P}}^*$ as the input data in the MLEM and KEM algorithms to iteratively reconstruct the dynamic PET images $\boldsymbol{X}$.

*MLEM algorithm* [26]:

$$\boldsymbol{X}^{n+1} = \boldsymbol{X}^n \circ (\boldsymbol{H}^T(\hat{\boldsymbol{P}}^* \oslash (\boldsymbol{HX}^n + \boldsymbol{S}))) \oslash (\boldsymbol{H}^T\boldsymbol{1}_{MN}) \tag{12}$$

where $\boldsymbol{X}^n$ is the estimate of image matrix $\boldsymbol{X}$ at the $n$th iteration, $\boldsymbol{1}_{MN}$ is an $M \times N$ all 1

matrix, $\circ$ is Hadamard product and $\oslash$ is Hadamard division, defined as $A \circ B = [a_{ij} b_{ij}]$ and $A \oslash B = [a_{ij}/b_{ij}]$ for the matrices $A = [a_{ij}]$ and $B = [b_{ij}]$ of the same dimension.

*KEM algorithm* [10]:

$$\mathbf{\Lambda}^{n+1} = \mathbf{\Lambda}^n \circ (\mathbf{K}^T \mathbf{H}^T (\hat{\mathbf{P}}^* \oslash (\mathbf{HK\Lambda}^n + \mathbf{S}))) \oslash (\mathbf{K}^T \mathbf{H}^T \mathbf{1}_{MN}), \tag{13}$$

$$\mathbf{X}^{n^*} = \mathbf{K\Lambda}^{n^*} \tag{14}$$

where $\mathbf{K} \in \mathcal{R}^{M_1 \times M_1}$ is a kernel matrix, $\mathbf{\Lambda} \in \mathcal{R}^{M_1 \times N}$ is the coefficient matrix of the image matrix $\mathbf{X}$ under the kernel matrix $\mathbf{K}$, satisfying $\mathbf{X} = \mathbf{K\Lambda}$, $\mathbf{\Lambda}^n$ is the estimate of $\mathbf{\Lambda}$ at the last iteration $n^*$, and $\mathbf{X}^{n^*}$ is the estimate of image matrix $\mathbf{X}$ at the last iteration $n^*$. The kernel matrix $\mathbf{K} = [k_{ij}]$ is given by

$$k_{ij} = \begin{cases} k(\tilde{\mathbf{x}}_i, \tilde{\mathbf{x}}_j), & j \in \mathcal{J} \\ 0, & \text{otherwise} \end{cases}, \quad i,j = 1, 2, \cdots, M_1, \tag{15}$$

where $k(\cdot, \cdot)$ is a kernel function such as the Gaussian kernel in (1), $\tilde{\mathbf{x}}_j$ is the prior image data of the voxels in the $j$th row of $\mathbf{X}$, $\mathcal{J}$ is the neighborhood.

The pseudo-code of Steps 1–3 is given in Algorithm 1 below.

**Algorithm 1** Pseudo-code of the proposed method

```
Require: Dynamic PET sinograms data matrix P; imaging system matrix H.
Ensure: Reconstructed dynamic PET image matrix X.
1: Extract kernel principal components Y of P using (1)-(4);
2: Compute adjacency matrix A using (5) and (6);
3: Compute Markov transition matrix F using A and (7);
4: Compute the optimal order m* using (9) to obtain the optimal kernel
graph filter Fᵐ*;
5: Filter sinogram data P using the optimal kernel graph filter Fᵐ* and
(10) to obtain denoised sinogram data P̂*.
6: Reconstruct PET images X using denoised sinogram data P̂* in MLEM
(12) or KEM (13) and (14).
```

*Remark* 1: By the kernel PCA theory [22, 24], the Eq (4) used in Step 1 is equivalent to

$$y_{li} = \sum_{j=1}^{N} \alpha_{lj} \hat{C}_{ij} = \phi(\mathbf{p}_i)^T \mathbf{v}_l = \phi(\mathbf{p}_i)^T \sum_{j=1}^{N} \alpha_{lj} \phi(\mathbf{p}_j),$$

where $\mathbf{v}_l = \sum_{j=1}^{N} \alpha_{lj} \phi(\mathbf{p}_j)$ is the $l$th eigenvector of the underlying covariance matrix $\mathbf{Q} = \frac{1}{N} \sum_{i=1}^{N} \phi(\mathbf{p}_i) \phi(\mathbf{p}_i)^T$. Hence, $y_{li} = \phi(\mathbf{p}_i)^T \mathbf{v}_l = \mathbf{v}_l^T \phi(\mathbf{p}_i)$ is the projection of $\phi(\mathbf{p}_i)$ on $\mathbf{v}_l$, which is used in conventional non-kernel PCA to extract the principal components $y_{li}$'s [22]. The Eq (4) avoids direct computation of $\mathbf{Q}$ and $\phi(\mathbf{p}_i)^T \mathbf{v}_l$ which is infeasible in kernel PCA, since $\mathbf{Q}$, $\mathbf{v}_l$ and $\phi(\mathbf{p}_i)$ are infinite dimensional and $\phi(\mathbf{p}_i)$ is not known explicitly [24].

*Remark* 2: The formula (6) for determining the number of the nearest neighbors, $k_i$, is derived as follows. Being the weighted edge between the $i$th and $j$th vertices of the graph, the $a_{ij}$ computed in (5) is a measure of the energy variation between $\mathbf{y}_i$ and its neighbor $\mathbf{y}_j$, which is determined by the energy variation between the original sinograms $\mathbf{p}_i$ and $\mathbf{p}_j$. As discussed in the Noisy dynamic sinograms subsection, at smaller $i$'s (early frames), the energy of $\mathbf{p}_i$ is low and energy variations between $\mathbf{p}_i$ and its neighbor sinograms are large. Hence, we need to use smaller $k_i$ to include fewer neighbour sinograms, ie fewer nonzero $a_{ij}$'s, to avoid the over-influence of later frames with high energy. Whereas, at larger $i$'s (later frames), the energy of $\mathbf{p}_i$ is high and energy variations between $\mathbf{p}_i$ and its neighbor sinograms are small, we need to use

larger $k_i$ to include more neighbour sinograms, ie more nonzero $a_{ij}$'s, to extract the similar structure information of time frames.

To determine $k_i$ adaptively for each $\boldsymbol{p}_i$, we use its 1-norm, $\|\boldsymbol{p}_i\|_1$, as the energy measure. From the discussion in the Noisy dynamic sinograms subsection, $\|\boldsymbol{p}_i\|_1$ satisfies $0 < \|\boldsymbol{p}_i\|_1 \leq \|\boldsymbol{p}_{i+1}\|_1 \leq \|\boldsymbol{p}_N\|_1$ and $0 < \|\boldsymbol{p}_i\|_1/\|\boldsymbol{p}_N\|_1 \leq \|\boldsymbol{p}_{i+1}\|_1/\|\boldsymbol{p}_N\|_1 \leq 1$ for $1 \leq i \leq N - 1$. Hence, $N\|\boldsymbol{p}_i\|_1/\|\boldsymbol{p}_N\|_1$, satisfying $0 < N\|\boldsymbol{p}_i\|_1/\|\boldsymbol{p}_N\|_1 \leq N$, gives the fraction of the $N$ frames of sinograms that should be used as the nearest neighbors for $\boldsymbol{p}_i$. Rounding the fraction to the nearest integer by the operator $r(\cdot)$, $k_i = r(N\|\boldsymbol{p}_i\|_1/\|\boldsymbol{p}_N\|_1)$ gives the integer number of nearest neighbors for each $\boldsymbol{p}_i$ with $1 \leq i \leq N$.

*Remark* 3: In (8), $\boldsymbol{F}^m$ attenuates the high frequency graph spectra of the noise and enhances the correlation (similarity) of the sinograms. In principle, the higher the order $m$, the stronger the correlation enhancement on the graph. But an overly high order $m$ may make the sinogram frames over-correlated and hence annihilate their dissimilarity. To avoid this, in our method, we use (9) to determine the optimal order $m^*$.

*Remark* 4: The kernel graph filter $\boldsymbol{F}^{m^*}$ used in (10) stems from the Gaussian kernel (1). Associated with the Gaussian kernel is an implicit nonlinear mapping $\phi(\boldsymbol{p}_i)$. This mapping can be made explicit and linear by using $\phi(\boldsymbol{p}_i) = \boldsymbol{I}\boldsymbol{p}_i$, with $\boldsymbol{I}$ an $M \times M$ identify matrix. In this case $\phi(\boldsymbol{p}_i) = \boldsymbol{p}_i$ and the kernel matrix becomes $\boldsymbol{C}_L = [c_{Lij}]$ with

$$c_{Lij} = \langle \boldsymbol{p}_i, \boldsymbol{p}_j \rangle = \boldsymbol{p}_i^T \boldsymbol{p}_j, \;\; i, j = 1, 2, \cdots, N, \tag{16}$$

which is a linear kernel. Replacing the nonlinear kernel matrix $\boldsymbol{C}$ used in Step 1 with this linear kernel matrix $\boldsymbol{C}_L$, we can also extract the principal components of sinograms and use them in Step 2 to obtain an optimal graph filter $\boldsymbol{F}_L^{m^*}$. Replacing $\boldsymbol{F}^{m^*}$ with $\boldsymbol{F}_L^{m^*}$ in (10), we can get graph filtered sinograms $\hat{\boldsymbol{P}}_L^*$. But $\boldsymbol{F}_L^{m^*}$ is not a kernel graph filter and its filtering is not on a kernel graph, because the principal components extracted using $\boldsymbol{C}_L$ are in the original sinogram space and do not carry the information about the infinite dimensional kernel space features of the sinograms. This affects the performance of $\boldsymbol{F}_L^{m^*}$ in sinogram denoising as shown in Result section.

## Results

To test the proposed KGF method and compare with the existing sinogram denoising methods, we have applied KGF, BM3D [14], SDIGF [15] and GF methods to the simulated and in vivo dynamic PET sinogram datasets to assess and compare their performance. As discussed before, BM3D and SDIGF are two popular sinogram denoising methods in the literature, and GF is a non-kernelized alternative graph filtering method to KGF. The assessment and comparison of these four methods on the same datasets was to demonstrate the advantage of KGF over the existing methods in sinogram denoising and enhancement of PET image quality, and to show the importance of kernel in the proposed graph filtering of dynamic sinograms.

### Dynamic PET data simulation

The anatomical model in BrainWeb dataset [28, 29] was used to simulate dynamic PET images. This anatomical model (Fig 1(a)) consists of 11 categories: Background, CSF, Grey Matter, White Matter, Fat, Muscle/Skin, Skin, Skull, Glial Matter, Connective, and Lesion. The time activity curve (TAC) was generated using three-compartment model [30]. The kinetic parameters were from [31]: $K_1 = 0.116 \; ml/g/min$, $k_2 = 0.254 \; min^{-1}$, $k_3 = 0.116 \; min^{-1}$ and $k_4 = 0.011 \; min^{-1}$ for gray matter, $K_1 = 0.059 \; ml/g/min$, $k_2 = 0.149 \; min^{-1}$, $k_3 = 0.090 \; min^{-1}$ and $k_4 = 0.013 \; min^{-1}$ for white matter, $K_1 = 0.089 \; ml/g/min$, $k_2 = 0.269 \; min^{-1}$, $k_3 = 0.135 \; min^{-1}$ and

$k_4 = 0.015 \, min^{-1}$ for lesion. Then, TAC in Fig 1(b) was filled into corresponding tissues to produce the dynamic noiseless ground truth images of the size $217 \times 181$. The simulated dynamic PET activity images consisted of 24 frames at the times: $4 \times 20s$, $4 \times 40s$, $4 \times 60s$, $4 \times 180s$ and $8 \times 300s$. The simulated PET activity image at the end of scan is shown in Fig 1(c). The bright spot in Fig 1(a) and 1(c) is a lesion simulated and inserted by BrainWeb.

These dynamic PET activity images were projected to the noiseless ground truth sinograms with 249 bins and 210 angles, via Fessler toolbox (github.com/JeffFessler/mirt). Then the 20% random events and the Poisson noise were simulated and added to the noiseless sinograms to produce noisy sinograms. The expected events number was set to 10 million over 60 minutes. Since Poisson process was a random process, ten noisy realizations was simulated and used in the simulation studies of the four denoising methods.

## Experiment setup and performance metrics

The KGF, BM3D, SDIGF and GF methods were used to denoise the noisy sinograms obtained from the simulated dynamic PET data described above. The denoised sinograms from each method were used in MLEM (12) and KEM (13) algorithms to reconstruct the PET dynamic images of each method. The correction factors for attenuation were estimated and used in both reconstruction methods to enhance performance. The following metrics were used in performance assessment and comparison.

Mean squared error (MSE):

$$\text{MSE}(\hat{\boldsymbol{x}}_j, \boldsymbol{x}_j) = 10 log_{10}\left(\frac{\sum_i (\hat{x}_{ij} - x_{ij})^2}{\sum_i (x_{ij})^2}\right)(dB) \tag{17}$$

where $\hat{x}_{ij}$ is the $i$th element of the $j$th frame of reconstructed PET image, and $x_{ij}$ is the $i$th element of the $j$th frame of ground truth. MSE is a measure of the difference between the reconstructed image and the ground truth image. The smaller the MSE, the closer the reconstructed image to the ground truth image and hence the better quality of reconstructed PET image.

Regional time activity (RTA):

$$\text{RTA}(\boldsymbol{x}_j^r) = \frac{1}{D}\sum_i |x_{ij}^r|, \tag{18}$$

where $\boldsymbol{x}_j^r$ is the vector of all pixels in a region of the $j$th frame of dynamic PET images, $x_{ij}^r$ is the $i$th element of $\boldsymbol{x}_j^r$, and $D$ is the dimension of $\boldsymbol{x}_j^r$. RTA measures PET image uptake in the regions of interest, such as the legion, gray matter and white matter of the brain, at an image frame. RTA is often indexed by the times of image frames and used to approx the time activity curve (TAC) of a region.

Regional mean absolute error (MAE) of PET time activity:

$$\text{MAE}(\hat{\boldsymbol{X}}, \boldsymbol{X}) = \frac{1}{N}\sum_{j=0}^{N-1} |\text{RTA}(\hat{\boldsymbol{x}}_j^r) - \text{RTA}(\boldsymbol{x}_j^r)| \tag{19}$$

where $\text{RTA}(\hat{\boldsymbol{x}}_j^r)$ and $\text{RTA}(\boldsymbol{x}_j^r)$ are respectively the RTAs of the reconstructed and the ground truth images of the $j$th frame. The MAE gives the average distance between the regional RTAs of the reconstructed image and the ground truth image. So it is a measure of PET image quantification in the regions of interest. The smaller the MAE, the closer the RTAs of the reconstructed and ground truth images, and hence the better quantification of reconstructed PET image.

The MSE, RTA and MAE above were evaluated on the PET images reconstructed by MLEM or KEM algorithms. The tuning parameters of these algorithms are dependent on the sinograms data used in image reconstruction. For fair comparison, we tuned the parameters of each method on the sinogram data to attain its lowest sinogram MSE value, and used the corresponding parameters for the method. Using this tuning method we found the best parameters for each evaluated method. For BM3D, the standard deviation of noise $\sigma$ was 20. For SDIGF, the windows of radius was 10, and the smooth parameter was 0.5. For GF, the $\epsilon$ in (9) was $1 \times 10^{-3}$, the number of principal component $d$ was 10, the $\sigma_2$ of Gaussian kernel function in (5) was 1. In KGF method, the number of kernel principal component $d$ in (4) was 7, the $\epsilon$ in (9) was $1 \times 10^{-3}$, the $\sigma_1$ of Gaussian kernel function in (1) was 0.5, the $\sigma_2$ of Gaussian kernel function in (5) was 1. The total iteration number of reconstruction methods was set to 100.

## Sinogram denoising results of different methods

Fig 2 compares the denoising results of the four methods at the low count Frame 8 (count = 68k), the middle count Frame 16 (count = 492 k) and the high count Frame 24 (count = 914k). The Noiseless column shows the noiseless ground truth sinograms, the Noisy column shows the noisy sinograms, and the other four columns show the sinograms denoised by BM3D, SDIGF, GF and KGF methods, respectively. As seen from the figure, KGF can effectively reduce the random noise in sinograms, achieving the lowest MSE at all count levels among the four methods. Whereas, the other three methods are effective only at the middle and high count levels and are ineffective at the low count level with much higher MSEs.

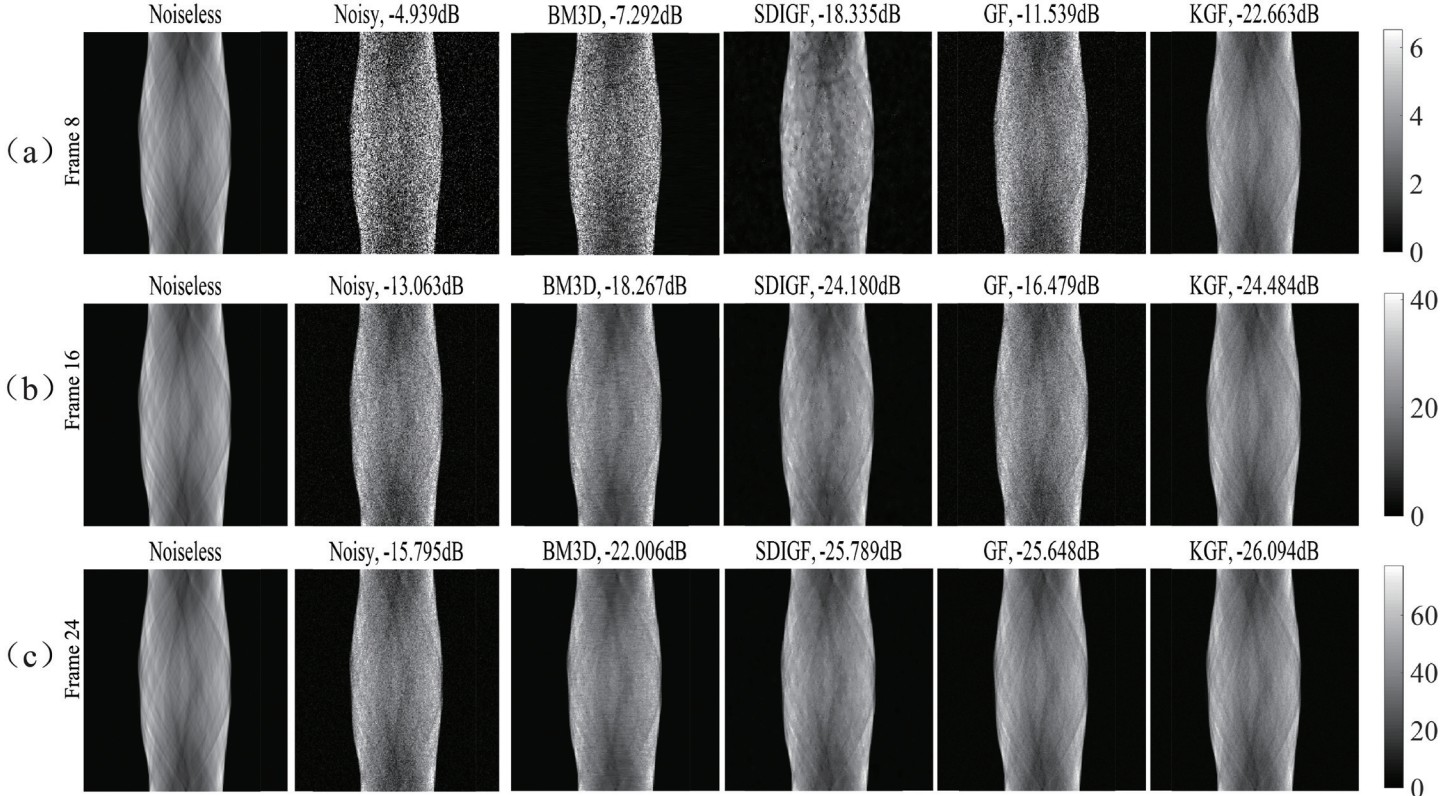

**Fig 2. Sinograms denoised by different methods: (a) Frame 8 (count = 68k), (b) Frame 16 (count = 492k), (c) Frame 24 (count = 914k).** Noiseless column: noiseless sinograms; Noisy column: noisy sinograms; other columns: sinograms denoised by respective methods. The dB values are MSEs of noisy or denoised sinograms.

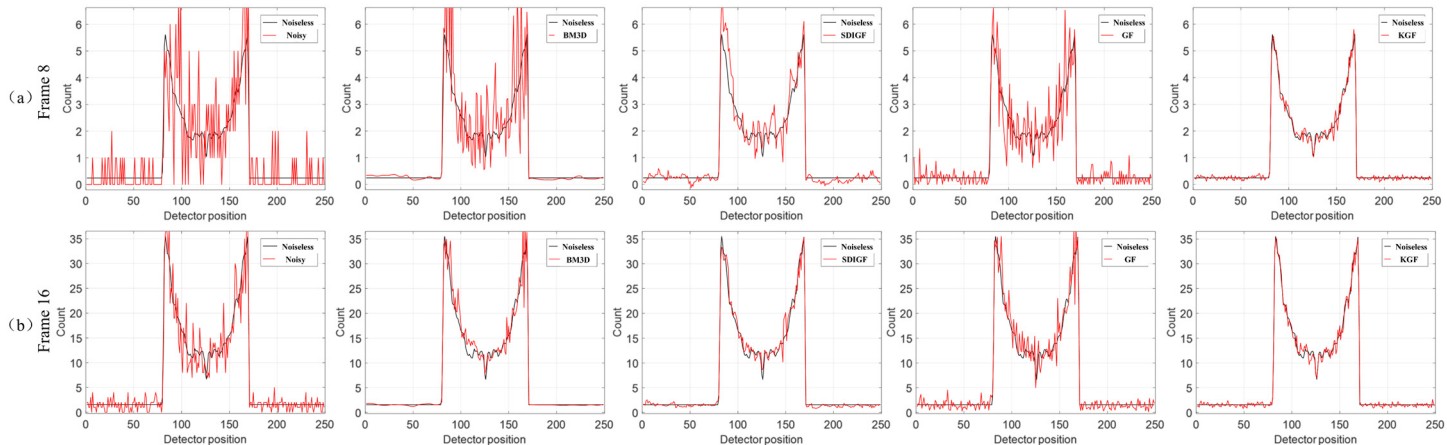

**Fig 3. Single angle curves of the sinograms denoised by different methods for Frame 8 and Frame 16.** Black line is noiseless sionogram cruve. The columns correspond to noisy sinogram and sinograms denoised by BM3D, SDIGF, GF and KGF (left to right).

Further, the results given in the last two columns of Fig 2 clearly show that KGF outperforms GF in sinogram denoising, achieving much lower MSE at all count levels, especially at the low and middle count levels. This demonstrates the advantage of KGF over GF and the benefit of using kernel in sinogram graph filtering.

Fig 3 compares the single angle curves of the four differently denoised sinograms and the noiseless sinograms. It can be seen from the plots that the sinograms denoised by KGF is closer to the noiseless ground truth sinograms in the low count Frame 8 and the middle count Frame 16. The curves at high count frames are not compared here because the denoising results of all methods are similar.

## MLEM reconstructed images from different denoising methods

Fig 4 compares the images reconstructed by MLEM algorithm from the corresponding sinograms shown in Fig 2. The Ground truth column shows the images from the noiseless ground true sinograms, the Nois column shows the images from the noisy sinograms, and the other four columns show the images from the sinograms denoised by BM3D, SDIGF, GF and KGF methods, respectively.

Fig 4(a) compares the reconstructed images at the low count Frame 8. It can be seen that the KGF image is closest to the ground truth image as compared with all other images, having the lowest MSE, lower noise, clearer texture, higher intensity (brightness) in the lesion area. The better quality of KGF image is the result of lower MSE of the KGF denoised sinograms shown in Fig 2(a).

Fig 4(b) compares the reconstructed images at the middle count Frame 16. The images from noisy and denoised sinograms are all better than at the low count Frame 8, since the count now is about seven times higher. But the KGF image is still the best among all these images, with the lowest MSE, more pronounced intensity in the lesion area, higher texture contrast and lower noise. These are due to the lower MSE of the KGF denoised sinograms shown in Fig 2(b).

Fig 4(c) compares the reconstructed images at the high count Frame 24. Because of the high count, the images from noisy and denoised sinograms are all much better than at the low

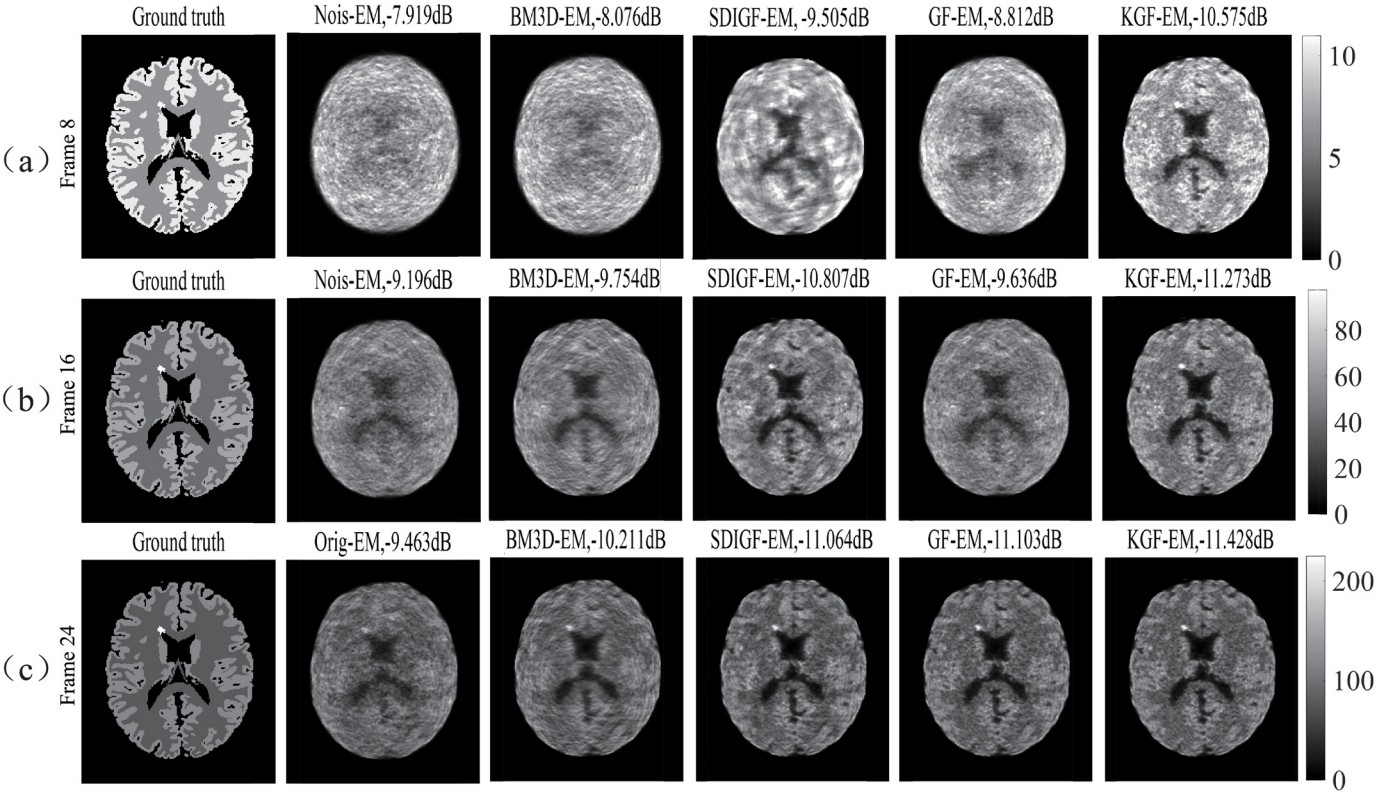

**Fig 4. MLEM reconstructed images from the corresponding sinograms shown in Fig 2.** (a) Frame 8 (count = 68k), (b) Frame 16 (count = 492k), (c) Frame 24 (count = 914k). Ground truth column: images from noiseless sinograms, Nois column: images from noisy sinograms, other columns: images from sinograms denoised by respective methods. The dB values are MSEs of reconstructed images.

count and middle count frames. But KGF image still has the lowest MSE and sharper textures due to the lower MSE of the KGF denoised sinograms shown in Fig 2(c).

Further, the images given in the last two columns of Fig 4 clearly show that KGF images are better than GF images at all count levels, especially at the low and middle count levels, with lower MSE, more pronounced intensity in the lesion area, higher contrast, sharper texture and lower noise. This demonstrates the advantage of KGF over GF and the benefit of using kernel in enhancing dynamic PET image quality and quantification.

Fig 5 compares the average MSEs (AMSEs) of all frames of MLEM reconstructed images from the sinograms denoised by the four methods. The average is taken over the ten realizations of Poisson noise. It can be seen that KGF images attain the lowest AMSE at all the frames.

Fig 6 compares the RTAs of MLEM reconstructed image frames from the noisy sinograms and the denoised sinograms of the four methods. The RTAs of lesion, gray matter and white matter regions are shown and indexed by the times of the 24 image frames, and are used to plot the local TACs in each region and compared with the TACs of ground truth image. Fig 6(a)–6(c) present the RTAs over the entire time range of 3,600 seconds, and Fig 6(d)–6(f) present the RTAs in the first 600 seconds. As seen from the plots, the TACs of KGF images are the closest to the TACs of ground truth image in all regions, especially in the early times shown in Fig 6(d)–6(f). Moreover, the local TACs of KGF images are smoother than those of BM3D, SDIGF and GF images because of its temporal filtering of dynamic sinograms. These are the desired properties in clinic applications. Table 1 lists the corresponding MAEs of the image

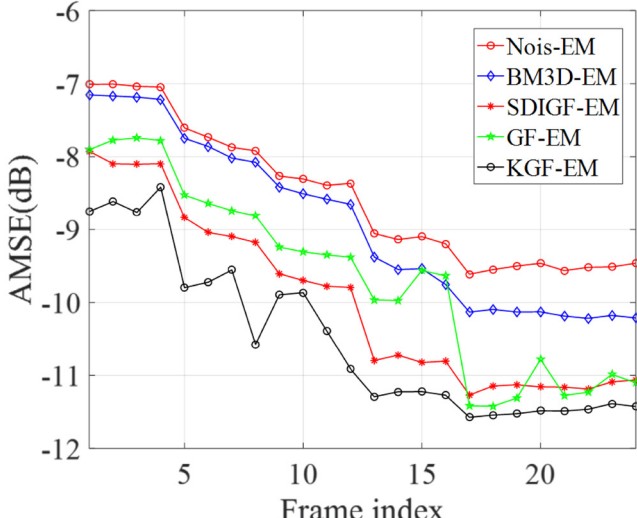

**Fig 5. AMSEs of MLEM reconstructed image frames from the sinograms denoised by different methods.**

**Fig 6. RTAs and local TACs of MLEM reconstructed image frames from the sinograms denoised by different methods.** (a) Lesion, (b) gray matter, and (c) white matter over 0–3600 seconds. (d) Lesion, (e) gray matter, and (f) white matter over 0–600 seconds.

**Table 1. MAEs of MLEM reconstructed images from noisy and denoised sinogram datasets.**

| Regions | Nois-EM | BM3D-EM | SDIGF-EM | GF-EM | KGF-EM |
|---|---|---|---|---|---|
| Lesion | 40.90 | 37.15 | 27.87 | 32.57 | **26.38** |
| Gray matter | 8.51 | 8.54 | 7.25 | 7.41 | **6.78** |
| White matter | 1.93 | 1.89 | 1.25 | 1.39 | **1.15** |

frames compared in Fig 6. As seen from the table, the KGF images attain the lowest MAEs in all regions as compared with those of other denoising methods.

## KEM reconstructed images from different denoising methods

As seen from the Nois column of Fig 4, the PET images reconstructed by MLEM from noisy sinogram data have poor spatial resolution and contrast. This is an inherent problem of MLEM with sinogram raw data, especially at low count [32]. Therefore, post-filtering is generally needed for MLEM images. The recently proposed KEM reconstruction method [10] uses kernelized MLEM to mitigate this problem and has shown promising results. To further verify the efficacy of our KGF denoising method, we tested it with KEM reconstruction method. The results are presented below.

Fig 7 shows KEM reconstructed low count (Frame 8), middle count (Frame 16) and high count (Frame 24) images of the same sinogram datasets shown in Fig 2. Compared with their

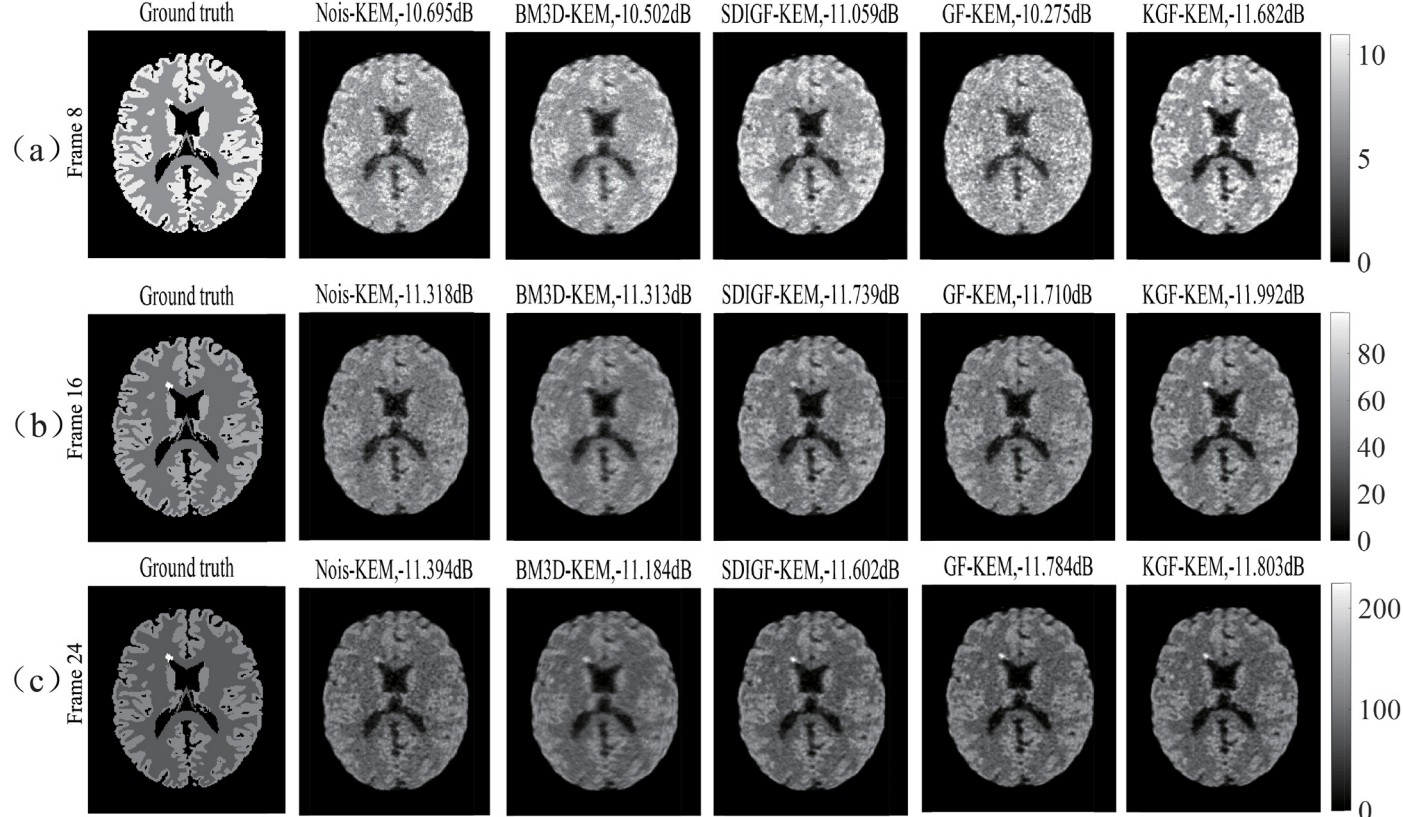

**Fig 7. KEM reconstructed images from corresponding sinogram datasets shown in Fig 2.** (a) Frame 8, (b) Frame 16, and (c) Frame 24. The dB values are MSEs of reconstructed images.

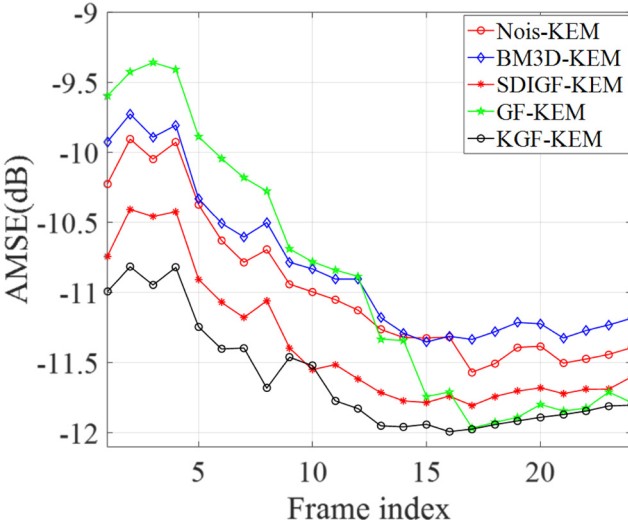

**Fig 8. AMSEs of KEM reconstructed image frames from the sinograms denoised by different methods.**

corresponding MLEM reconstructed images shown in Fig 4, the KEM reconstructed images of all datasets have shown lower MSE, and finer texture at all count levels. Among these images, the KGF images attain the lowest MSEs and the highest uptake in the lesion region at the low and middle count frames.

The images given in the last two columns of Fig 7 show that under KEM reconstruction, the KGF images are still better than GF images at all count levels, especially at the low and middle count levels, with lower MSE, more pronounced intensity in the lesion area, higher contrast, sharper texture and lower noise. This demonstrates again the advantage of KGF over GF and the benefit of using kernel in enhancing dynamic PET image quality and quantification.

Fig 8 compares the average MSEs (AMSEs) of all frames of KEM reconstructed images from the sinograms denoised by the four methods. The average is taken over the ten realizations of Poisson noise. The same as MLEM reconstruction case, the KGF images reconstructed by KEM attain the lowest AMSE at all the frames.

Fig 9 compares the RTAs of KEM reconstructed image frames from the noisy sinograms and the denoised sinograms of the four methods. The plots and comparisons are the same as those of Fig 6, and the results are almost the same. Once again, the RTAs and TACs of KGF images are the closest to those of ground truth image in all regions.

Table 2 lists the corresponding MAEs of the KEM reconstructed image frames compared in Fig 9. The same as MLEM reconstruction case, the KGF images reconstructed by KEM attain the lowest MAEs in all regions as compared with the images of other denoising methods.

## Influence of KGF parameters

Fig 10(a) plots the influence of the number of kernel principal component $d$ in (4) on average MSE of all sinogram frames. Noise in sinograms influences the robustness and reliability of adjacency matrix. In general, the kernel principal components in the eigenvector directions of small eigenvalues always contain noise. Hence, we alternatively choose the eigenvectors of larger eigenvalues to obtain the kernel principal components so as to eliminate the interference of noise. By this method, we find that $4 \leq d \leq 14$ provides good results.

Fig 10(b) shows the influence of $\sigma_2$ of Gaussian kernel function in (5) on on average MSE of all sinogram frames. $\sigma_2$ can be viewed as a measure of similarity between graph vertices. If $\sigma_2$ is

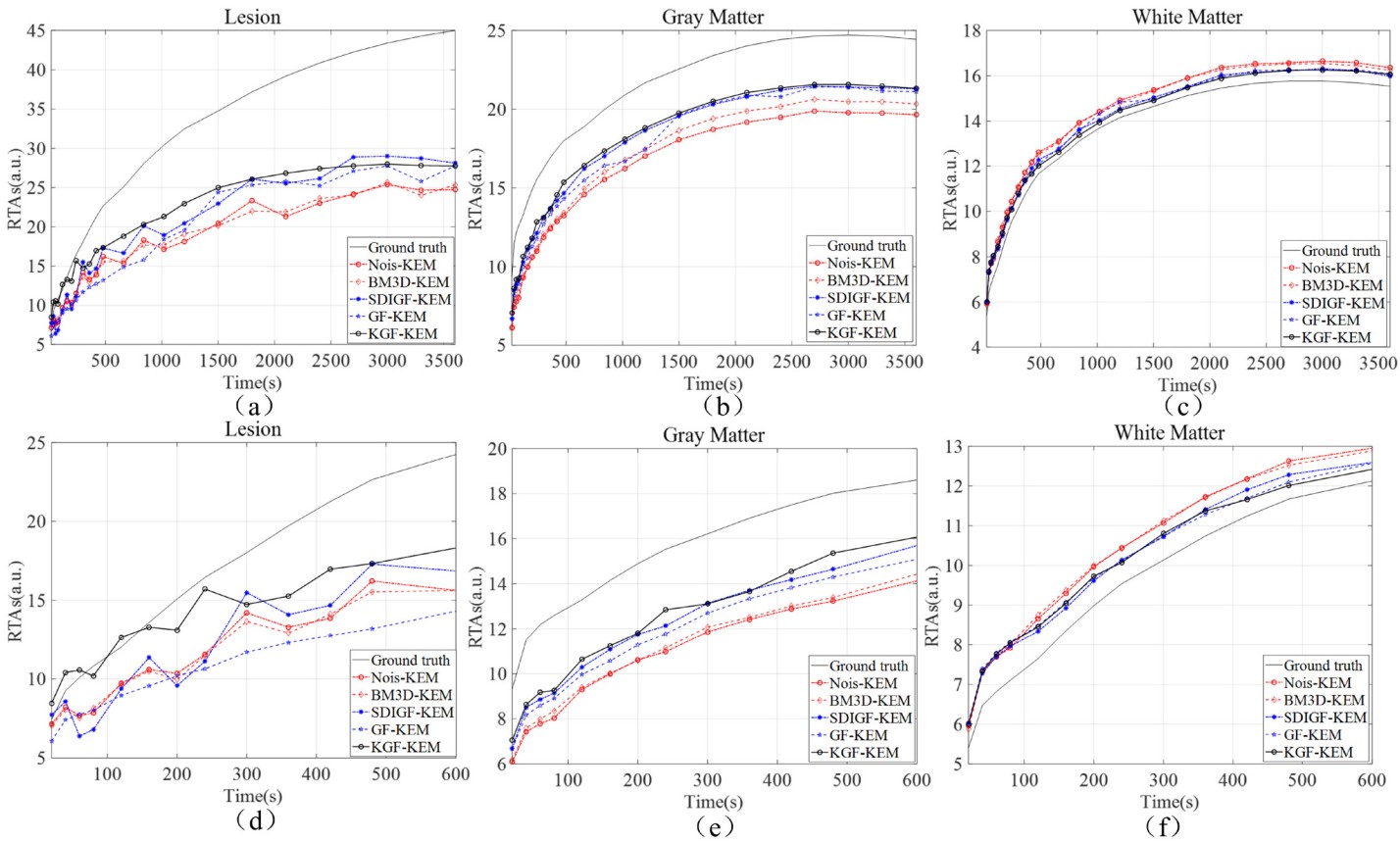

**Fig 9. RTAs and local TACs of KEM reconstructed image frames from the sinograms denoised by different methods.** (a) Lesion, (b) gray matter, and (c) white matter over 0–3600 seconds. (d) Lesion, (e) gray matter, and (f) white matter over 0–600 seconds.

too small, the graph becomes disconnected and less effective at suppressing noise. As $\sigma_2$ grows larger, the adjacency matrix become smoother and becomes effective at noise reduction. But, overly large $\sigma_2$ results in the loss of resolution and texture structure in the constructed images. The KGF yields a relatively stable MSE when $0.5 \leq \sigma_2 \leq 5$.

Fig 10(c) shows the influence of the constant $\epsilon$ in (9) on sinograms' MSE. In our algorithm, $\epsilon$ is used to compute the optimal order of low-pass filter $m^*$. The smaller the $\epsilon$, the higher the order $m^*$. As we described in Methods section, the higher the order $m$, the stronger the correlation enhancement on the graph. But an overly high order $m^*$ may make the sinogram frames over-correlated and hence annihilate their dissimilarity. In Frame 8, since TAC varies rapidly, the high order over enhances the similarity between frames, making KGF unable to reduce the noise effectively. Hence, in our paper, we use $10^{-4} \leq \epsilon \leq 10^{-1}$.

Since KGF is insensitive to $\sigma_1$ of Gaussian kernel function in (1) in simulations, we do not discuss its influence here.

**Table 2. MAEs of KEM reconstructed images from noisy and denoised sinogram datasets.**

| Regions | Nois-EM | BM3D-EM | SDIGF-EM | GF-EM | KGF-EM |
|---|---|---|---|---|---|
| Lesion | 36.55 | 36.55 | 29.63 | 32.41 | **27.96** |
| Gray matter | 11.69 | 10.27 | 7.81 | 8.53 | **7.36** |
| White matter | 2.07 | 1.92 | 1.21 | 1.28 | **1.07** |

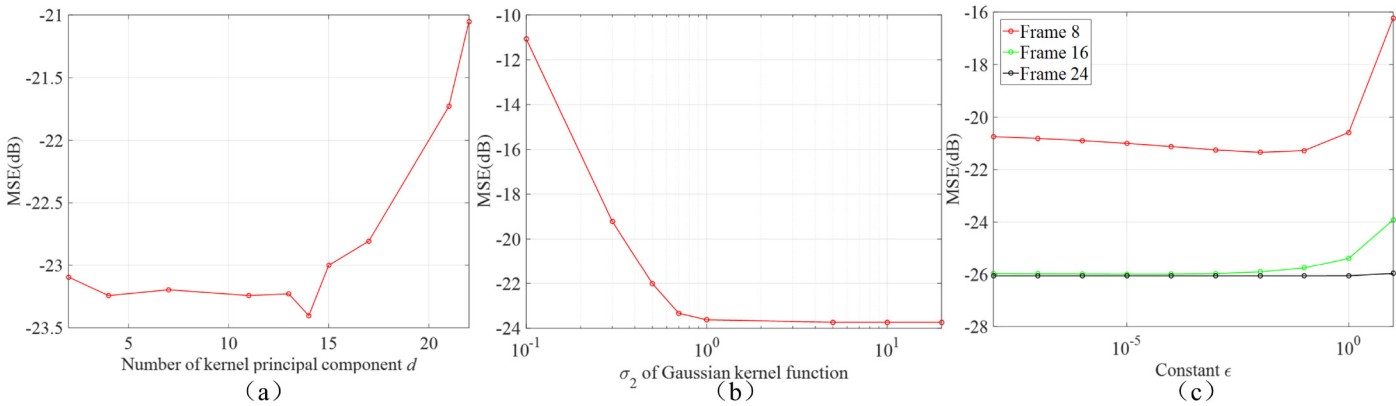

**Fig 10. The influence of parameters in KGF (a) number of kernel principal component $d$ and (b) $\sigma_2$ of Gaussian kernel function and (c) constant $\epsilon$ on sinograms MSE.**

## Tests on real life in-vivo data

To validate the simulation results and test the proposed method in real life application, we performed similar experiments on in-vivo PET data. The in-vivo PET data was from [33], which was acquired from volunteers with 90 min on a Siemens (Erlangen) Biograph 3 Tesla molecular MR (mMR) scanner, with an average dose 233 MBq of [18-F] FDG infused into the subject over the course of the scan at a rate of 36mL/hr. The raw data was rebinned into 26 sinograms frames with 360 angular projections, 319 radial bins, and attenuation correction by pseudoCT method. For BM3D, $\sigma$ value is set to 25. For SDIGF, the windows of radius is set to 6, and the smooth parameter is set to 0.1. For GF, $\epsilon$ in (9) is set to $1 \times 10^{-4}$, the number of principal components is set to 10, the $\sigma_2$ of Gaussian kernel in (5) is set to 0.5. For KGF, $\epsilon$ in (9) is set to $1 \times 10^{-4}$, the number of kernel principal components $d$ in (4) is set to 8, the $\sigma_1$ of Gaussian kernel in (1) is set to 0.15 and the $\sigma_2$ of Gaussian kernel in (5) is set to 0.5. The total iteration number of reconstruction methods is set to 50.

Similar to simulation experiment, we first compare the sinogram denoising results of BM3D, SDIGF, GF and KGF for the low count Frame 4 (scan duration 24 seconds), middle count Frame 9 (scan duration 54 seconds) and high count Frame 14 (scan duration 84 seconds). Next, we compare the images reconstructed by MLEM and KEM using the denoised sinograms of these frames. Since the ground truth noiseless sinograms and images are unknown for real PET scan data, only qualitative visual comparisons are presented.

Fig 11 compares the original sinograms and the sinograms denoised by different methods. As seen from the figure, the original sinograms are quite noisy and KGF denoising makes the sinograms much smoother and less noisy at all count levels. Whereas, the other denoising methods can only do so at the high count Frame 14, and their effect is limited at the low and middle count Frames 4 and 9.

Fig 12 shows the MLEM reconstructed images from the original and denoised sinograms. It can be seen that at the high count Frame 14, all denoising methods can effectively reduce image noise without compromising the main texture features, while KGF denoising attains the best visual effect. However, at the low and middle count Frames 4 and 9, only KGF can achieve these and the effect of other methods are rather limited.

Fig 13 shows the KEM reconstructed images from the original and denoised sinograms. Clearly, the image quality of all denoising methods is significantly improved by KEM

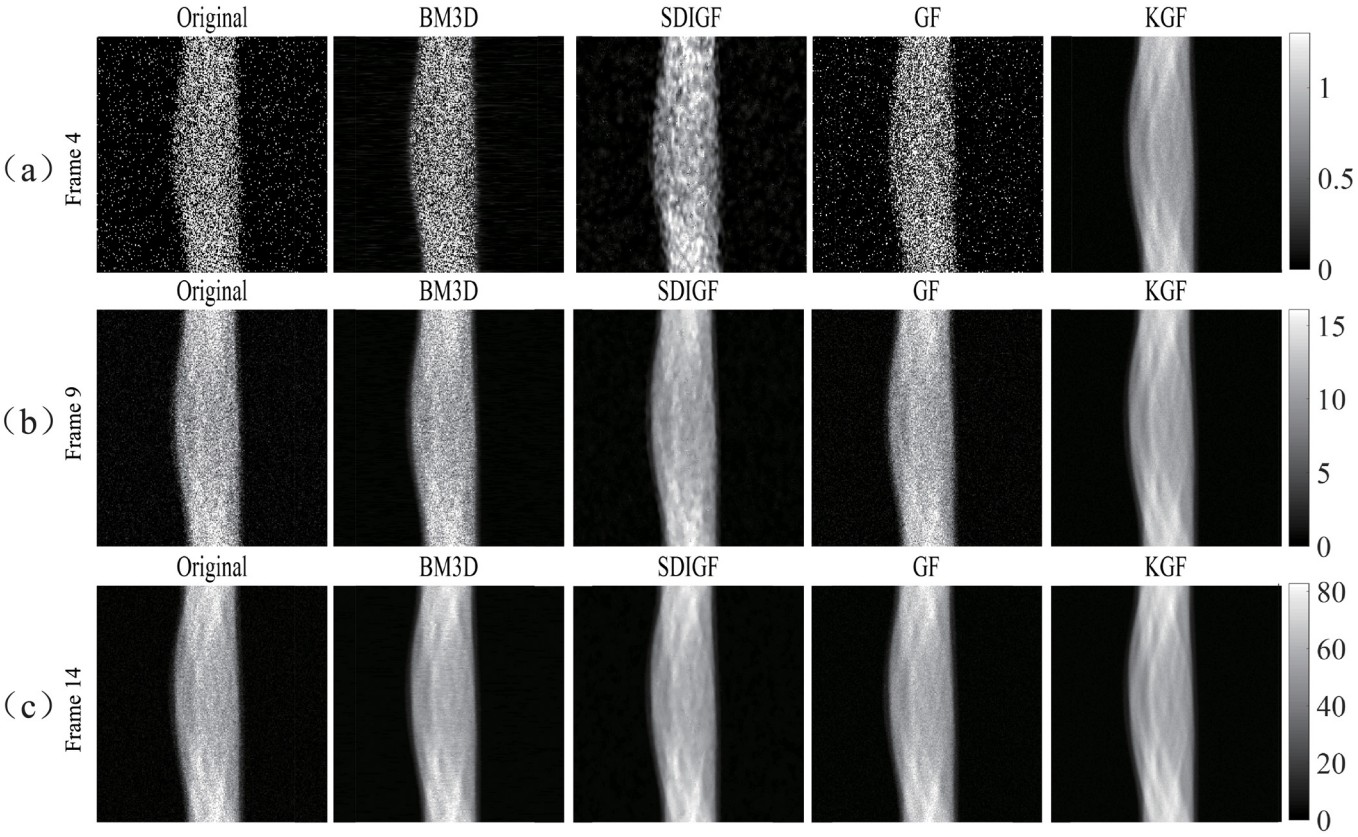

**Fig 11. In vivo sinograms denoised by different methods for (a) low count frame 4, (b) middle count frame 9, (c) high count frame 14.** Original column: original sinograms, other columns: sinograms denoised by respective methods.

reconstruction at all count levels. But the quality of KGF images is still the best at each count level and more consistent across all levels.

## Discussion

As discussed in Methods section, KGF replaces the original sinograms $p_i$'s with their weighted averages $\hat{p}_i^{*}$'s. The weighting is along the columns of the optimal kernel graph filter $F^{m^*}$, and the weighting coefficients are learned adaptively from the sinograms' kernel principal components. The average is across the sinogram time frames and is repeated $m^*$ times when the optimal kernel graph filter's order $m^* > 1$. This amounts to the $m$th order adaptive lowpass graph spectrum filtering in the temporal direction of the sinograms, which makes KGF more flexible and more effective than the existing sinogram denoising methods in attenuating the noise while preserving the image information. These have been proven by simulation studies and in-vivo data tests.

The SDIGF method also uses the temporal information to denoise sinograms. However, it simply rebins all sinogram time frames into one high count frame and uses this high count frame to guide all the time frames. This simplistic method does not explore the sophisticated relationships between the sinogram frames, and hence its efficacy is limited, especially in the low count and middle count frames as shown in Figs 2(a), 2(b), 11(a) and 11(b).

Radial Gaussian kernel and kernel PCA of dynamic PET sinograms are two of the key ingredients in the derivation of KGF. Gaussian kernel manifests dynamic PET sinograms in

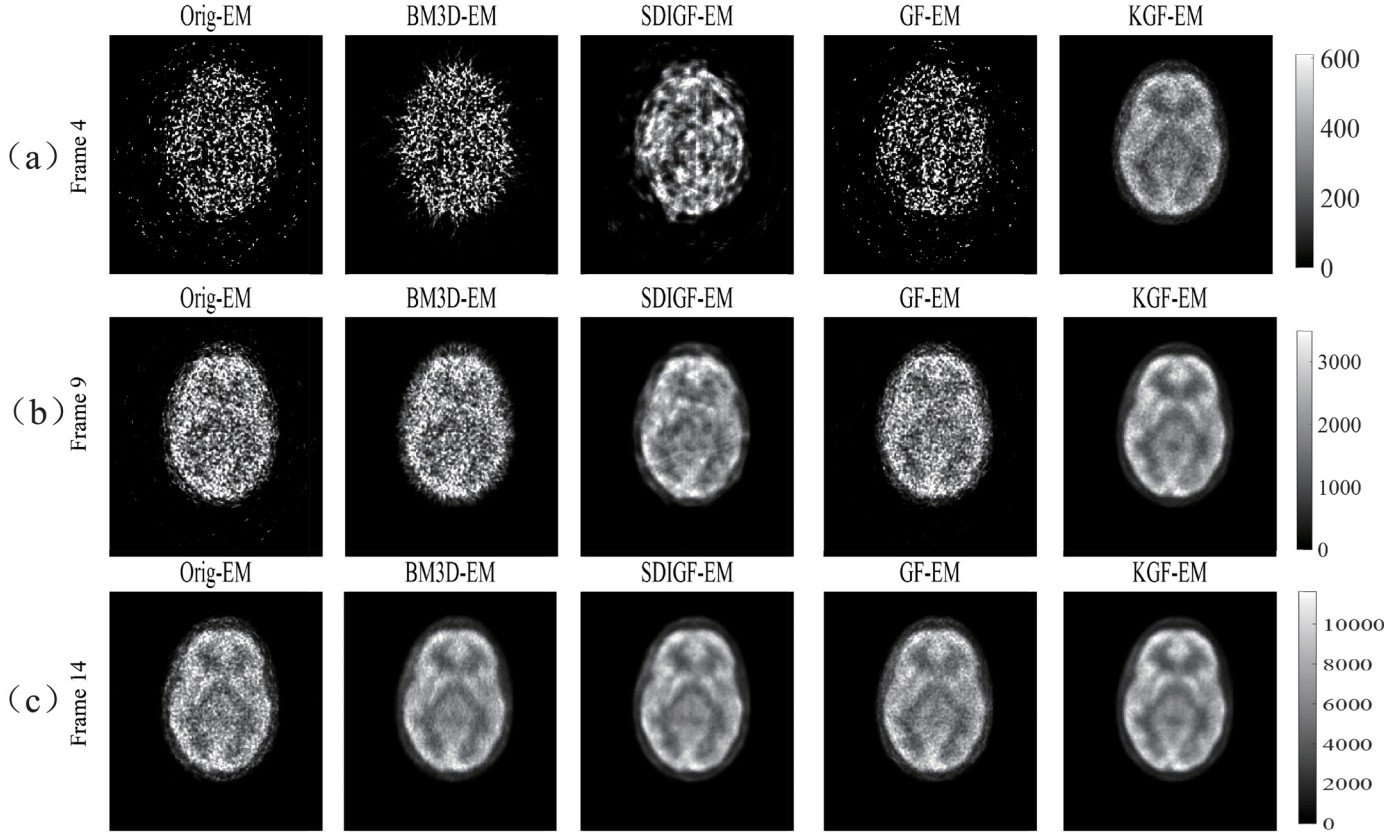

**Fig 12. MLEM reconstructed images from the in-vivo sinogram datasets shown in Fig 11.** Column Orig: original dataset, other columns: datasets denoised by respective methods.

the infinite dimensional feature space, and kernel PCA extracts the nonlinear principal components of sinograms from such feature space. Therefore, the KGF derived from the nonlinear principal components better adapts to the dynamic PET sinograms data and outperforms the non-kernelized GF method in sinogram denoising and image quality enhancement. These have been confirmed by simulation studies and in-vivo data tests. Since KGF use kernel PCA tool, it need more time to calculate kernel projection. But it is acceptable. In the simulation experiment, the total time of kernel PCA is 0.23s, and PCA need to spend 0.03s. Those are run in Matlab version R2016a on a Windows 10 PC with 3.6 GHz Intel Core i7–7700 CPU.

As seen from Figs 4 and 7 and Tables 1 and 2, the KEM reconstructed images attain lower MSE and clearer background as compared with MLEM reconstructed images, especially in the low count frame. But their textures are somewhat smoothed out and the uptake in lesion region becomes lower. This is because that KEM improves the contrast to noise ratio at the cost of possible over-smoothing the features unique to PET data [34]. Comparing the images of Fig 7, it can be seen that the KGF denoised sinograms can mitigate the over-smoothness of KEM to a certain extent, with enhanced contrast and clearer texture in the images. It is therefore fair to say that the KGF denoised sinograms can significantly improve the image quality of KEM and MLEM reconstructions, especially at the low count frames.

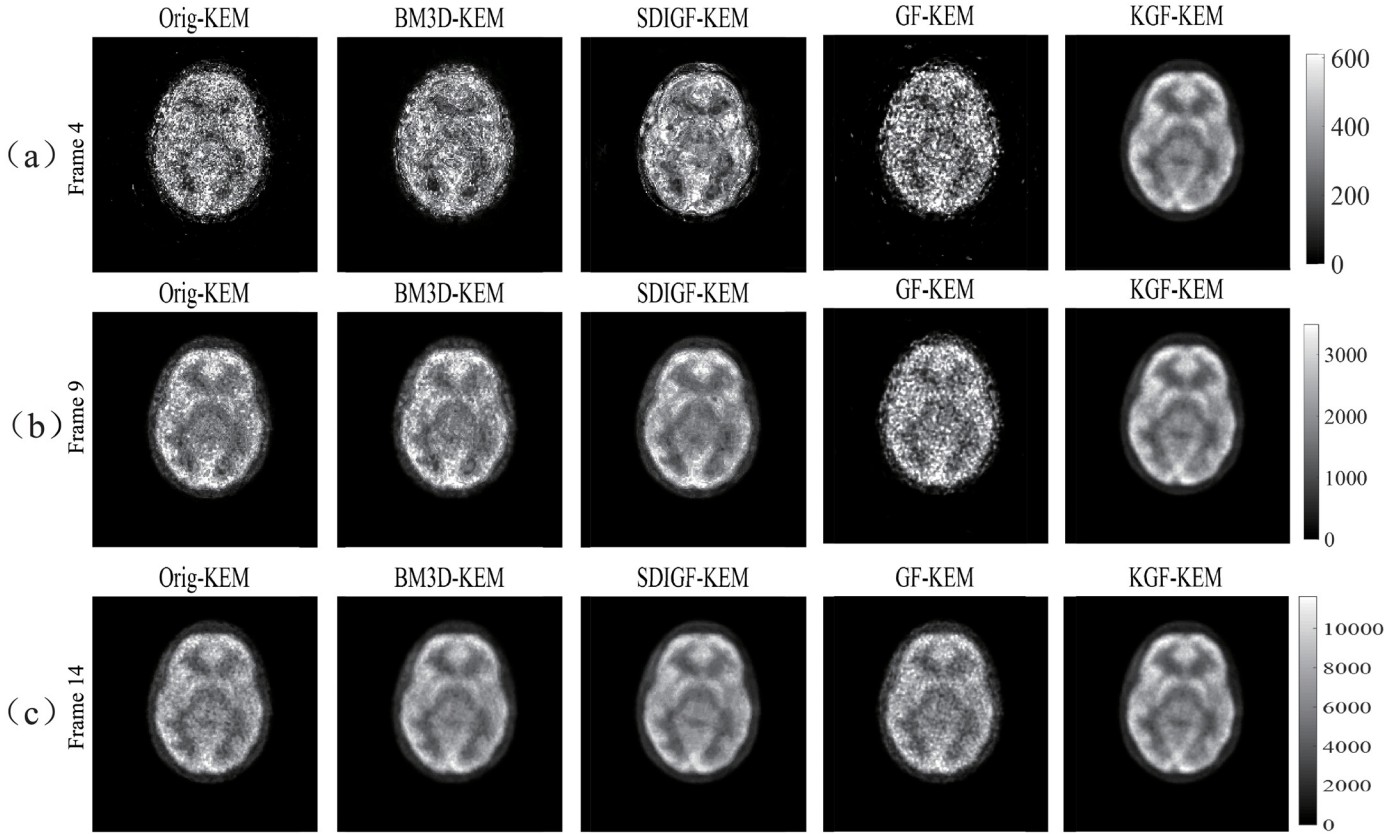

**Fig 13. KEM reconstructed images from the in-vivo sinogram datasets shown in Fig 11.** Column Orig: original dataset, other columns: datasets denoised by respective methods.

## Conclusion

A novel kernel graph filtering method has been proposed to effectively reduce the noise in dynamic PET sinograms. It can be used with the existing methods for sinogram based dynamic PET image reconstruction to improve the quality of reconstructed image. Extensive simulation studies and tests on in-vivo dynamic PET data have shown the efficacy and advantages of the proposed method over the existing methods in sinogram denoising and image enhancement of dynamic PET, especially at low count.

The new method is derived by entangling the technical tools of kernel, kernel PCA, kernel signals on graph, graph filter construction, and graph spectrum filtering. This approach has opened a new avenue for developing new methods for PET image enhancement.

## Supporting information

**S1 Data.**
(XLSX)

## Author Contributions

**Conceptualization:** Shiyao Guo, Jingxin Zhang.

**Data curation:** Shiyao Guo, Jingxin Zhang.

**Formal analysis:** Shiyao Guo, Yuxia Sheng, Jingxin Zhang.

**Funding acquisition:** Yuxia Sheng, Li Chai.

**Investigation:** Shiyao Guo, Li Chai, Jingxin Zhang.

**Methodology:** Shiyao Guo, Yuxia Sheng.

**Project administration:** Shiyao Guo, Yuxia Sheng, Li Chai.

**Resources:** Shiyao Guo, Yuxia Sheng, Li Chai, Jingxin Zhang.

**Software:** Shiyao Guo.

**Supervision:** Yuxia Sheng, Li Chai, Jingxin Zhang.

**Validation:** Shiyao Guo.

**Visualization:** Shiyao Guo, Li Chai.

**Writing – original draft:** Shiyao Guo, Yuxia Sheng, Jingxin Zhang.

**Writing – review & editing:** Shiyao Guo, Yuxia Sheng, Li Chai, Jingxin Zhang.

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
