## [Decision Letter · Decision Letter 0]

17 Sep 2021

PONE-D-21-19088Kernel graph filtering - a new method for dynamic sinogram denoisingPLOS ONE

Dear Dr. Sheng,

Thank you for submitting your manuscript to PLOS ONE. After careful consideration, we feel that it has merit but does not fully meet PLOS ONE’s publication criteria as it currently stands. Therefore, we invite you to submit a revised version of the manuscript that addresses the points raised during the review process.

We look forward to receiving your revised manuscript.

Kind regards,

Mohammadreza Hadizadeh

Academic Editor

PLOS ONE

A clean copy of the edited manuscript (uploaded as the new *manuscript* file).

Additional Editor Comments (if provided):

Reviewers' comments:

Reviewer's Responses to Questions

**Comments to the Author**

1. Is the manuscript technically sound, and do the data support the conclusions?

Reviewer #1: Partly

2. Has the statistical analysis been performed appropriately and rigorously? 

Reviewer #1: Yes

3. Have the authors made all data underlying the findings in their manuscript fully available?

Reviewer #1: Yes

4. Is the manuscript presented in an intelligible fashion and written in standard English?

Reviewer #1: Yes

5. Review Comments to the Author

Reviewer #1: Dear authors:

The denoising of PET data has always been the center of attention in the field of medical imaging. There are many studies in the current literature regarding this issue. In this paper, the authors introduced the kernel graph filtering method for denoising PET sinograms. The idea behind this work is interesting. However, the explanation of the theory employed in this method is confusing to me. Some of the parameters are not defined properly and I cannot go through the derivations carefully. The authors need to re-write these once again with clear explanation. I attached my detailed comments. Considering my comments, I recommend major revision of the manuscript before being considered for publication in PLOS ONE.

6. PLOS authors have the option to publish the peer review history of their article (what does this mean?). If published, this will include your full peer review and any attached files.

Reviewer #1: No

---

## [Author Response · Author response to Decision Letter 0]

29 Oct 2021

Response to comments of the editor and reviewers is presented in attach file (Response to Reviewers.pdf).

---

## [Decision Letter · Decision Letter 1]

9 Nov 2021

Kernel graph filtering - a new method for dynamic sinogram denoising

PONE-D-21-19088R1

Dear Dr. Sheng,

We’re pleased to inform you that your manuscript has been judged scientifically suitable for publication and will be formally accepted for publication once it meets all outstanding technical requirements.

Kind regards,

Mohammadreza Hadizadeh

Academic Editor

PLOS ONE

Reviewers' comments:

Reviewer's Responses to Questions

**Comments to the Author**

1. If the authors have adequately addressed your comments raised in a previous round of review and you feel that this manuscript is now acceptable for publication, you may indicate that here to bypass the “Comments to the Author” section, enter your conflict of interest statement in the “Confidential to Editor” section, and submit your "Accept" recommendation.

Reviewer #1: All comments have been addressed

2. Is the manuscript technically sound, and do the data support the conclusions?

Reviewer #1: Yes

3. Has the statistical analysis been performed appropriately and rigorously? 

Reviewer #1: Yes

4. Have the authors made all data underlying the findings in their manuscript fully available?

Reviewer #1: Yes

5. Is the manuscript presented in an intelligible fashion and written in standard English?

Reviewer #1: Yes

6. Review Comments to the Author

Reviewer #1: Authors addressed all my comments very well. The paper has been improved significantly. Please review your paper one more time for few small punctuation and grammatical issues. I congratulate the authors for developing such interesting method. I have no other comments on the present paper, and therefore I recommend the paper to be accepted for publication.

7. PLOS authors have the option to publish the peer review history of their article (what does this mean?). If published, this will include your full peer review and any attached files.

Reviewer #1: No

---

## [Editor Report · Acceptance letter]

19 Nov 2021

PONE-D-21-19088R1 

Kernel graph filtering – a new method for dynamic sinogram denoising 

Dear Dr. Sheng:

I'm pleased to inform you that your manuscript has been deemed suitable for publication in PLOS ONE. Congratulations! Your manuscript is now with our production department. 

Kind regards, 

on behalf of

Dr. Mohammadreza Hadizadeh 

Academic Editor

PLOS ONE